# Comprehensive Transcriptome Analysis of Responses during Cold Stress in Wheat (*Triticum aestivum* L.)

**DOI:** 10.3390/genes14040844

**Published:** 2023-03-31

**Authors:** Lei Li, Chenglin Han, Jinwei Yang, Zhiqiang Tian, Ruyun Jiang, Fei Yang, Kemeng Jiao, Menglei Qi, Lili Liu, Baozhu Zhang, Jishan Niu, Yumei Jiang, Yongchun Li, Jun Yin

**Affiliations:** 1National Key Laboratory of Wheat and Maize Crop Science, College of Agronomy, Henan Agricultural University, Zhengzhou 450046, China; 2National Engineering Research Centre for Wheat/Henan Technology Innovation Centre of Wheat, Henan Agricultural University, Zhengzhou 450046, China

**Keywords:** transcriptome, wheat, cold stress, freezing tolerance

## Abstract

Wheat production is often impacted by pre-winter freezing damage and cold spells in later spring. To study the influences of cold stress on wheat seedlings, unstressed Jing 841 was sampled once at the seedling stage, followed by 4 °C stress treatment for 30 days and once every 10 days. A total of 12,926 differentially expressed genes (DEGs) were identified from the transcriptome. *K*-means cluster analysis found a group of genes related to the glutamate metabolism pathway, and many genes belonging to the bHLH, MYB, NAC, WRKY, and ERF transcription factor families were highly expressed. Starch and sucrose metabolism, glutathione metabolism, and plant hormone signal transduction pathways were found. Weighted Gene Co-Expression Network Analysis (WGCNA) identified several key genes involved in the development of seedlings under cold stress. The cluster tree diagram showed seven different modules marked with different colors. The blue module had the highest correlation coefficient for the samples treated with cold stress for 30 days, and most genes in this module were rich in glutathione metabolism (ko00480). A total of eight DEGs were validated using quantitative real-time PCR. Overall, this study provides new insights into the physiological metabolic pathways and gene changes in a cold stress transcriptome, and it has a potential significance for improving freezing tolerance in wheat.

## 1. Introduction

Wheat (*Triticum aestivum* L.) is one of the world’s most important food crops and possesses enormous economic significance [1]. Under climate change, cold events are likely to continue or even become more serious [2]. To survive, grow, and reproduce in the fluctuating environment yearly, plants have evolved a number of complex mechanisms to respond to abiotic stresses. The transcription of stress genes regulates the adaptation process, including developmental, physiological, and biochemical changes [3,4,5]. The stress genes are currently known to encode two main categories of proteins: one category is mainly involved in plant protection, while the other category is involved in regulating the expression of downstream genes. This second category contains transcription factors (TFs) that can bind to the promoter regions of stress-responsive genes, thereby improving stress tolerance in plants [6,7,8]. Through the analysis of stress response promoters, many *cis*-acting and *trans*-acting elements involved in the transcription of stress response genes have been identified [4,9].

Temperature change is a major seasonal change in a temperate climate. For plants, it is very important to have a mechanism to survive the freezing temperatures that occur in winter. Abiotic stress factors, such as cold and frost, severely affect the growth and development of plants, including wheat, ultimately affecting crop yields [10]. The freezing resistance of plants refers to their ability to survive and continue to function after being exposed to low temperatures. Plants do not usually show freezing resistance in the warm growing season, but with the change in seasons, plants perceive the temperature drop in winter and thus develop freezing resistance [11,12]. The freezing resistance of plants is a complex process involving multiple genes [13]. The crown of wheat is a key organ for survival through the winter under low-temperature stress. Previous studies revealed that the vascular transition zone shows increased sensitivity to freezing compared to the apical meristem [14]. The response of cereal crops to low temperatures in winter undergoes two processes. However, the relationship between the cold stress response and vernalization in plants remains elusive [15].

Most temperate plants exhibit chilling tolerance. While many species display constitutive freezing tolerance, some species require cold acclimation [16]. Many plants have increased their frost resistance at low non-freezing temperatures, which is a phenomenon called cold acclimation [11]. Cold acclimation is a complicated process that aims to improve plants’ frost resistance and the survival rate of plants under an unfavorable low temperature. In laboratory experiments, a low temperature induced plants to positively adapt, which was related to the increased accumulation of several stress proteins. A short period of cold acclimation can effectively improve plant resistance to low-temperature stress. Typically, one or two days of low non-freezing temperatures is enough for cold acclimation for most plant species to adapt to cold [11,17]. This relatively fast freezing tolerance (FT) is necessary to cope with sudden temperatures. The low-temperature (LT) tolerance of grain largely relies on a highly integrated inducible gene system. Studies have shown that the complete expression of the LT tolerance gene only occurs in the vegetative stage, while the cold acclimation ability of plants in the reproductive stage is limited [18].

Plant hormones are essential for regulating freezing resistance. Jasmonic acid (JA) has been reported to positively regulate the ICE-CBF pathway to enhance the freezing resistance of *Arabidopsis thaliana* [19]. Hindering JA biosynthesis and signal transduction produces hypersensitivity to freezing stress [19]. The ethylene (ET) signaling pathway is known to negatively regulate FT. EIN3 inhibits the expression of *CBFs* and type-A *ARR* genes by directly binding their promoters [20,21]. At low non-freezing temperatures, many plant species can acquire an increased low-temperature tolerance [22]. This process involves multiple changes in gene expression, membrane lipid composition, compatible penetration (such as proline), antioxidant levels, and other traits [23]. Molecular analysis has identified low-temperature response genes, such as dehydrin and crystallization genes [11]. Between 0 °C and 10 °C, a cold adaptation response occurs; it affects protein and DNA structure and photosynthetic activity [24,25]. The main regulators of the cold adaptation response are C-repeat (CCGAC) binding factors (CBFs) [26,27] and bZIP (leucine zipper domain) TFs [28]. Their roles support the synthesis of compatible solutes, cold storage proteins, and fatty acid desaturases, which can stabilize cell viability and increase resistance to low-temperature and freezing stress [29,30].

Plant hormones also play critical roles in the cold acclimation. ET has been reported to accumulate in *Arabidopsis*, bean (*Phaseolus Vulgaris*), winter rape (*Brassicanapus*), tomato (*Solanum Lycopersicum*), and winter rye (*Secale cereal*) after exposure to low temperatures [31,32,33,34,35]. Exposure to cold temperatures for several days improves cold tolerance and usually leads to a higher tolerance to temperatures below zero (freezing tolerance) [16]. Transcriptome studies have shown that many genes are up-regulated early after being transferred to a low-temperature environment and may contribute to combating the effects of oxidative stress [36]. Despite the fact that there are similarities in the response mechanisms of plants to cold stress, the molecular mechanisms of wheat facing cold stress are unique and complex due to different varietal characteristics, and many questions remain to be clarified. This study reports the molecular mechanism of wheat Jing 841 under cold stress at the seedling stage. It attempts to elucidate the expression patterns, regulatory networks, and molecular mechanisms of important TFs and genes in the process of low-temperature stress in wheat, to provide a scientific basis for clarifying the molecular mechanism of cold tolerance in winter wheat, and to provide important target genes for molecular design and breeding.

## 2. Materials and Methods

### 2.1. Plant Materials and Cold Stress Treatment

The wheat Jing 841 used in this study is a winter variety [37]. The surface of seeds were disinfected with 75% ethanol for 1 min, then they were disinfected with 10% sodium hypochlorite for 15 min, and they were finally washed with sterile water 5 times. The seeds were then planted into Petri dishes with wet filter paper and placed in an incubator at 22 °C under 16 h light and 8 h darkness cycle. After 10 days, the seedlings were moved to another condition. The temperature of the cold stress treatment was kept at 4 °C, and the control condition was 22 °C.

### 2.2. Trait Measurements and Morphology Observations

The samples were observed while the length of the seedlings was about 8–10 cm, and the pictures of the seedlings were photographed using a digital camera (Nikon Coolpix 4500, Nikon Corporation, Tokyo, Japan).

### 2.3. Library Preparation, RNA Extraction, and Sequencing

The total RNA of the four samples was prepared using RNeasy Plant Mini Kit (QIAGEN, Hilden, Germany) according to the manufacturer’s protocol, including the control group (CK: CK0-1, CK0-2, and CK0-3), seedlings after cold stress treated for 10 days (D10: D10-1, D10-2, and D10-3), seedlings after cold stress treated for 20 days (D20: D20-1, D20-2, and D20-3), and seedlings after cold stress treated for 30 days (D30: D30-1, D30-2, and D30-3). Each sample had three biological replicates, and each sample contained more than 20 individuals. All the samples were immediately frozen in liquid nitrogen for the subsequent experiments. The RNA concentration and integrity were checked using the NanoDrop 2000 and the RNA Nano 6000 Assay Kit (NanoDrop Technologies, Wilmington, DE, USA). Qualified samples were used for library construction and Illumina sequencing.

The twelve cDNA libraries were sequenced using Illumina HiSeq X Ten from Biomarker Biotechnology Corporation (Beijing, China) [38].

### 2.4. Transcriptome Analyses

The clean reads were compared with the wheat reference genome IWGSC RefSeq v1.0 (https://urgi.versailles.inra.fr/download/iwgsc/IWGSC_RefSeq_Assemblies/v1.0/, 1 January 2023) using HISAT2 [39]. The unigenes were annotated based on Nr (NCBI non-redundant protein sequences); Nt (NCBI non-redundant nucleotide sequences). The mapped reads were pieced using StringTie software and compared with the original genome annotation information, in search of the original unannotated transcription regions. KEGG orthology-based annotation of the sequence’s genes was analyzed using KOBAS2.0 [40], and the amino acid sequences of the new genes were predicted. After predicting the amino acid sequences of the new genes, HMMER [41] software was used to compare with the Pfam database to complement and refine the original genome annotation information. The gene expression levels were quantified as fragments per kilobase of the transcript per million fragments mapped (FPKM) [42]. To identify differentially expressed genes (DEGs), the DEseq method was used with a false discovery rate (FDR) threshold of <0.01 and a fold change value (FC) threshold of ≥8 of significance for differential expression. Biological repetition correlation was evaluated using the Pearson correlation coefficient (PCC) and the principal components analysis (PCA) [43,44]. DEGs were screened with a threshold parameter of Log_2_FC ≥ 3 or ≤−3, and gene co-expression analysis was performed on the expression values using the *K*-means clustering algorithm [45]. A gene was considered valid gene if its average FPKM value among the twelve libraries was more than one. The statistical analyses of genes in different samples were performed using valid genes. All the analyses were conducted on BMKCloud (https://www.biocloud.net, accessed on 1 January 2023). The accession number of the transcriptome data in NCBI is PRJNA903114.

### 2.5. Identification of Transcription Factors, Transcription Repressors, and Kinases

The protein sequences of wheat were analyzed using the iTAK v1.6 (http://itak.feilab.net, accessed on 1 January 2023) to classify and identify TFs, transcription repressors (TRs), and kinases [46].

### 2.6. K-Means Clustering of DEGs

In this study, the expression values of three replicates of each sample were used for gene co-expression analysis using the *K*-means clustering algorithm [45]. A *K*-means clustering method was used to cluster the samples in Section 2.4, and the number of clusters was 8. We used log_2_(FPKM+1) and expression values normalized by the Z-score to create a heat map, and we used heat map in TBtools for the visualization.

### 2.7. Weighted Gene Co-Expression Network Analysis (WGCNA)

WGCNA was used to construct gene co-expression networks by analyzing pairwise correlations among genes with similar expression trends. Using the WGCNA tool, gene co-expression networks were constructed based on the pairwise correlations of gene co-expression trends in all the sampled tissues. The R package WGCNA converts adjacency into topological overlap, which measures the network connectivity of a gene by summing its adjacency with other genes. The hierarchical clustering was used to classify genes with similar expression profiles into modules based on TOM dissimilarity. At least 30 modules are used to create a gene dendrogram. Tangents were selected for merging modules by calculating the dissimilarity of the feature genes in each module. Hub genes, which are highly interconnected with nodes in a module, were considered functionally significant. Key genes were selected from the DEGs in each module, and the DEGs in the key modules were screened for further analysis [47,48].

### 2.8. qRT-PCR Validation

Four samples of Jing 841 with four treatments (CK, D10, D20, and D30) were prepared for qRT-PCR. Primers were designed using Primer Premier 5.0 software (www.premierbiosoft.com/primerdesign/index.html, accessed on 1 January 2023). The primer sequence is listed in Appendix A. The TranScript^®®^ integrated First-Strand cDNA Synthesis SuperMix for qPCR was used for reverse transcription (Transgenic Biotechnology, Beijing, China). The TransStart^®®^ Top Green qPCR SuperMix (2×) was employed for qRT-PCR in accordance with the manufacturer’s instructions on the CFX ConnectTM Real-Time System (Bio-Rad, Hercules, CA, USA). The configuration of qRT-PCR is 20 µL volumes [38]. The 2^−∆∆Ct^ method was utilized to calculate the gene expression level, using the wheat *actin* gene as an internal control [49].

## 3. Results

### 3.1. Transcriptome Data Overview

A total of 114.57 Gb clean data and about 382 million reads were obtained from the twelve libraries, averaging 8.59 Gb and 31 million reads per sample. The G + C content of all the samples varied from 56.13% to 57.97%, and the average Q30 percentage was 94.73% (Appendix A). The percentage of unique mapped reads also exceeded 94.70% (Appendix A). Based on the mapped results, 22,928 new genes were discovered, of which 15,242 have functional annotations (Appendix A).

A good correlation was clearly shown among the transcriptome sequencing samples, as shown in the three-dimensional map of PCA (Figure 1). The colored points on the PCA map represented the distinct samples, and the three eigenvectors (22.25%, 14.90%, 10.59%) distinctly separated them, which was consistent with the morphological pattern of the different tissues. Furthermore, PCC analysis demonstrated that each correlation coefficient exceeded 0.98 among all the replicated samples, indicating that the transcriptome sequencing reads were of a good quality for the subsequent identification of DEGs and regulatory network construction.

The transcriptional network of Jing 841 was established, and the key genes regulating cold stress treatment were identified. From the twelve libraries, 133,718 genes (unigenes) were obtained, and 125,405 unigenes were BLAST-annotated in multiple databases (Appendix A). In the four samples, 53,625, 51,516, 51,609, and 48,193 valid genes were identified (Figure 2a). The number of valid genes only expressed in CK, CK-D10, CK-D20, and CK-D30 was 4542, 452, 1357, and 3044, respectively (Figure 2b). The number of DEGs was the least between CK-D10, while the number of DEGs was the most between CK-D30.

### 3.2. Identification of DEGs

DEGs were identified through a pairwise comparison of the twelve libraries. When the filtering threshold remained unchanged at FDR < 0.01, the number of identified DEGs in the four tissues subjected to the four different treatments varied: 71,557, 33,231, 19,174, and 12,364. The number of DEGs decreased as the fold change increased. (Figure 2c).

To identify key genes related to cold stress development, significant DEGs (Log_2_FC ≥ 3 or ≤3) were screened by comparing samples treated with cold stress at different times with CK. In the sample pairs of CK-D10, CK-D20, and CK-D30, a total of 3108, 4197, and 11,869 DEGs (Appendix A and Figure 2c) were identified (FDR < 0.01 and FC ≥ 8). TF genes have long been an important research focus as key regulatory components in the regulation of gene expression. In this study, we annotated TFs based on specific DNA-binding domain signatures from the Plant-TFDB database (http://planttfdb.cbi.pku.edu.cn, accessed on 1 January 2023) using RNA-seq. We detected 854 unique differentially expressed TFs among the DEGs (Appendix A). It showed that the transcription of TFs is important in the wheat cold stress response. Among the TFs of differential genes during cold stress treatment for 10 days, the bHLH-, C2H2-, and MYB-related families seem to play a more important role in the cold response, because there were 21, 17, and 11 in these 3 families, respectively. Each member has a differential expression. Among the TFs of DEGs during the cold stress treatment, the bHLH, MYB, and NAC families may play a relatively important role in responding to cold stress because each of these 3 families have 73, 59, and 56 members with a differential expression (Figure 3).

DEGs were classified into 45 out of 52 subcategories using the GO database (Appendix A), while the KEGG database classified these genes into six major metabolic pathways.

In addition, the further enrichment and analysis of DEGs was performed. We present the top enrichment categories of the DEGs in biological processes, cellular components, and molecular functions in Table 1. In the comparison between the three groups of differently treated samples and the control group, the top enrichment terms of DEGs in the biological process, cell composition, and molecular function were the same in each group. They were nucleosome assembly (GO:0006334), nucleosome (GO:0000786), and protein heterodimerization activity (GO:0046982). The nucleosome assembly was very important to reconstruct the functional chromatin structure, which affected many biological processes, including DNA replication, repair, recombination, transcription, cell proliferation and differentiation, and individual development. These enrichment results suggested that most genes were associated with many biological processes, such as individual development.

The top metabolic pathways (Table 2) were classified based on gene annotation in the KEGG database. The DEGs of CK-D10 involved the top three enriched pathways that were photosynthesis-antenna proteins (ko00196), starch and sucrose metabolism (ko00500), and phenylpropanoid biosynthesis (ko00940). The DEGs of CK-D20, which involved the top three enriched pathways, were photosynthesis-antenna proteins (ko00196), DNA replication (ko03030), and benzoxazinoid biosynthesis (ko00402). The DEGs of CK-D30 involved the top three enriched pathways that were fatty acid degradation (ko00071), porphyrin and chlorophyll metabolism (ko00860), and starch and sucrose metabolism (ko00500). During this cold stress treatment, the reason for the increase in the starch and sucrose content was to better prevent freezing damage.

### 3.3. DEGs Co-Expression Clusters

To identify DEGs with similar expression patterns in response to cold stress, we performed *K*-means clustering tests for 12,926 DEGs (Appendix A) and generated 8 optimal clusters (Figure 4). Compared with the control CK, clusters K1, K5, and K3 contained 4043 genes, and their expression levels gradually increased with the accumulation of cold stress time. The expression of clusters K2 (687 genes) and K4 (2033 genes) with the accumulation of cold stress time showed a trend of first decreasing, then being gradual, and then decreasing. Cluster K6 (3825 genes) showed a gradual decrease in gene expression with the accumulation of cold stress time. The gene expression of cluster K7 (1166 genes) accumulated with the cold stress time showed a rapid increase at first, then gradually decreased in the next 20 days. Cluster 8 had 1172 genes, which decreased rapidly in the first 10 days of responding to cold stress and stabilized in the next 20 days.

To investigate the metabolic pathways in which the clustered genes may participate, several pathways were identified using the KEGG database (Appendix A and Figure 4b). Among the eight clusters, clusters K1 and K3 exhibited similar expression patterns and were found to be involved in glutathione metabolism (ko00480), aminoacyl-tRNA biosynthesis (ko00970), and tyrosine metabolism (ko00350). Meanwhile, clusters K2, K4, and K6 had the largest number of genes and similar expression patterns, and they were found to participate in starch and sucrose metabolism (ko00500) and phenylpropanoid biosynthesis (ko00940). Interestingly, the metabolism of starch and sucrose (ko00500) and phenylpropanoid biosynthesis (ko00940) were involved in six out of the eight clusters. Glutathione metabonomics (ko00480) in the eight clusters was involved in three clusters.

### 3.4. DEGs Co-Expression Network Analysis with WGCNA

The WGCNA was conducted on 12,927 differential genes (Appendix A). The cluster tree diagram showed seven different modules (marked with different colors), with each branch representing a module, and each leaf in the branch denoting a gene. The characteristic gene of a module is the first main component of a given module and can be regarded as a representative of the gene expression profile of the module. These seven modules were well correlated with the tissue expression profiles of their representative genes in the four samples (Figure 5b,c). The correlation coefficient between the blue module and the D30 sample was 0.99 (the error was 1 × 10^−9^), which indicated a high degree of correlation between the module and the sample. In addition, the correlation coefficient between the red module and the D10 sample was 0.95 (the error was 2 × 10^−6^), while the correlation coefficient between the green module and the D20 sample was 0.94 (the error was 5 × 10^−6^).

We enriched different modules in the KEGG database. The main pathways for genes in the turquoise module were DNA replication (ko03030), benzoxazinoid biosynthesis (ko00402), cyanoamino acid metabolism (ko00460), and phenylpropanoid biosynthesis (ko00940). The main ways for genes in the red module to be enriched were photosynthesis-antenna proteins (ko00196), circadian rhythm-plant (ko04712), and flavonoid biosynthesis (ko00941). The main pathways for genes in the green module were photosynthesis-antenna proteins (ko00196), plant hormone signal transduction (ko04075), and monoterpenoid biosynthesis (ko00902). The main pathways for genes in the blue module were glutathione metabolism (ko00480), cutin, suberin, and wax biosynthesis (ko00073), and aminoacyl-tRNA biosynthesis (ko00970).

### 3.5. Screening the Key Genes of Cold Treatment Changes with WGCNA

Complex traits in crops are typically controlled by multiple transcriptional networks. To identify the co-expression networks related to the cold resistance, we utilized the R WGCNA software, which was based on the FPKM and phenotypic data from the four treatments: before cold stress treatment, and after 10, 20, and 30 days of cold stress treatment, respectively. The clustering of the samples and correlation coefficients showed that the biological replicates were highly repeatable, and no outliers needed to be eliminated (Appendix A). We used the automatic R package blockwiseModules network construction method to identify co-expression modules (Appendix A). This allowed the module to be visualized with a color scheme that shows highly correlated genes in the same color and weakly correlated genes in a different color (Appendix A). The module construction process showed that the functional color modules were divided. After merging modules with similar expression patterns, it produced seven color modules, each of which consisted of genes with similar expression patterns (Figure 5b).

To identify hub genes in our interested modules, we selected the top 150 genes according to the kME value as hub genes in each module. In the 6618 genes of the turquoise module, based on the iTAK results, a total of 333 TFs, 64 TRs, and 257 kinase genes were identified. TraesCS7B01G016800 (UDP-Glc glucosyltransferase) was selected as the hub gene of the turquoise module. In the primary TraesCS7B01G016800 network, TraesCS7B01G016800 was directly co-expressed with 149 genes, including 2 TFs (belonging to the bHLH and HB-HD-ZIP families, respectively), 2 TRs, and 1 kinase genes (Figure 6a).

In the 233 genes of the red module, a total of 11 TFs, 4 TRs, and 2 kinase genes were uncovered. TraesCS6B01G383200 (cold shock protein CS66 in wheat, Gene Name = CS66) was selected as the hub gene of the red module. In the primary TraesCS6B01G383200 network, TraesCS6B01G383200 was co-expressed with 149 genes, including 7 TFs (bHLH and HSF families), 1 TR, and 2 kinase genes (Figure 6b).

In the 253 genes of the green module, 13 TFs, 1 TR, and 12 kinase genes were identified. TraesCS6B01G274200 (ERF034 in *Arabidopsis thaliana*) was selected as the hub gene of the green module. In the primary TraesCS6B01G274200 network, TraesCS6B01G274200 was directly co-expressed with 149 genes, including 8 TFs (bHLH, HSF, WRKY, and APETALA2/ETHYLENE RESPONSE FACTOR (AP2/ERF) families) and 8 kinase genes (Figure 6c).

The correlation coefficient of the blue module was the highest with the samples treated with cold stress for 30 days. Most genes in this module were enriched in glutathione metabolism (ko00480). In the 4076 genes of the blue module, a total of 215 TFs, 32 TRs, and 290 kinase genes were identified. TraesCS1D01G189600, TraesCS1D01G190100, TraesCS5B01G426300, TraesCS7D01G030800, and TraesCS7D01G050800 were selected as the hub genes of the red module. In the primary network, these hub genes belong to glutathione S-transferase. The co-expression network of the blue module comprised 150 genes, including 5 TFs (bHLH, MADS-MIKC, HB-HD-ZIP, MYB, and PLATZF families) and 1 kinase gene (Figure 6d).

The hub genes of each module were basically enriched in the key KEGG pathway of each module. The most interesting thing was that bHLH was involved in the expression of four important modules, which also showed that bHLH played important roles in the abiotic stress responses. There were also some key TFs involved in abiotic stress including WRKY and AP2/ERF-ERF TFs.

### 3.6. Validation of Expression Level of DEGs

To validate the results of our RNA-seq, eight DEGs were chosen for transcription validation using qRT-PCR. The relative expression levels of these eight key DEGs were analyzed using qRT-PCR across the four cold stress treatments of the seedlings to assess the accuracy of the RNA-seq results. The relative expression patterns of the tested DEGs were found to be positively correlated with the fold change variations obtained from the RNA-seq results (Figure 7). The correlation coefficients between qRT-PCR and RNA-seq for the 15 DEGs were observed to be within the range of 0.74 to 0.99. This indicated that our RNA-seq results were both reliable and accurate.

## 4. Discussion

The plant acclimatization response to different abiotic stresses involves complex molecular regulatory networks. For example, many isomers of plant thiol peroxidase play a central role in plant adaptation to stresses such as low temperatures, high amounts of light, or osmotic pressures [50]. In order to maintain plant life under stress conditions, antioxidant enzymes such as superoxide dismutase (SOD), catalase (CAT), peroxidase (POX), etc., play a crucial role [51]. This study reported an important economic gramineous wheat transcriptome of cold stress. High-quality transcription modules were achieved by sequencing with Super Illumina sequencing. Our present results can serve to not only enrich the existing transcriptome data resources for wheat but also offer valuable insights into the molecular mechanism of wheat cold stress resistance.

WGCNA showed that four expression modules had the highest correlation with cold stress. The combined analysis of WGCNA and *K*-means showed that five metabolic pathways were significantly associated with traits of interest, including phenylpropanoid biosynthesis (ko00940), benzoxazinoid biosynthesis (ko00402), flavonoid biosynthesis (ko00941), hormone signal transduction (ko04075), and glutathione metabolism (ko00480). This indicates that the transcription regulatory factors of related genes may be potential targets to improve the cold resistance of wheat crops.

The correlation coefficient of the blue module was the highest treated with cold stress for 30 days. The genes presented in the blue module exhibited a significant enrichment in glutathione metabolism (ko00480). Glutathione is one of the major endogenous antioxidants in plants, and plays a crucial role in plant defense mechanisms [52]. Glutathione S-transferase (GST), glutathione peroxidase (GPX), and glutathione reductase (GR) utilize glutathione to safeguard plants against abiotic stress [53]. Cold stress is known to produce reactive oxygen species (ROS) in plants and ROS accumulation may play a role in regulating cell death [54]. GST and GPX, as key enzymes of cellular detoxification systems, are essential in defending cells against ROS [55].

Another module that we were interested in was the green module. The genes present in the green module exhibited a significant enrichment in plant hormone signal transduction (ko04075) including the AP2/ERF TF family TF [3]. The ERF TF family plays a crucial role in regulating plant development, hormonal signaling, and environmental responses [56,57]. The AP2/ERF family is a large family of plant-specific TFs, which share a conserved DNA-binding domain [58]. It is worth noting that the regulation of cuticular wax accumulation is attributed to certain members of the AP2/ERF family that were originally isolated from *Arabidopsis*, namely *WAX INDUC-ER1/SHINE1* (*WIN1/SHN1*) [59,60]. Three *SHINE* genes *AtSHN1*, *AtSHN2*, and *AtSHN3* were classified into the ERF-B6 clade [57,61]. The overexpression of *SHINE1* (*SHN1*) in *Arabidopsis thaliana* increased the deposition of epidermal wax, increased the permeability of stratum corneum, decreased the stomatal index, and enhanced drought resistance [60]. Studies have indicated that *TaSHN1* may act as a positive regulator of drought stress tolerance in wheat, potentially by promoting an increase in alkane accumulation and a decrease in stomatal density. Therefore, *TaSHN1* holds promise as a valuable candidate gene for developing new wheat cultivars with improved drought tolerance [62]. The gene *WIN1* is a positive regulator of *Arabidopsis* epidermal wax biosynthesis. TraesCS6A01G181400 and its homologous sequence TraesCS6B01G210300 are homologous to *WIN1* in *Arabidopsis thaliana* and are differentially expressed in this module. Thus, we inferred that the regulatory network of the *WIN1* involved in the complex may have been evolutionarily conserved between *Arabidopsis* and wheat. Considering the differences between *Arabidopsis* and wheat, the conserved *WIN1* gene might be the best operable target locus in the breeding of stress-resistant varieties in wheat and related crops.

The cold shock gene was up-regulated during cold stress in wheat, which enhanced the cold adaptability of wheat. A cold shock affected the gene expression along with the starch and sucrose metabolic pathways. During cold adaptation, a metabolic readjustment occurs, which slows down growth and development and activates the defense process [63].

## 5. Conclusions

In this study, we identified 12,926 DEGs from transcriptome comparisons of seedling Jing 841 accessions with cold stress at four developmental stages. By studying the gene expression patterns and comparing the whole transcriptome in different cold stress periods and seedlings without cold stress treatment, it was determined that prolonged cold stress forced the suppression of plant metabolism, especially photosynthesis and sugar metabolism, and induced stress response genes, such as ERF and bHLH TFs. However, the regulation of *CBF* gene expression was not the only cold signaling factor. At the same time, it was also affected by various plant hormones, which indicated that CBFs were the center of the interaction between chilling and the hormone signaling pathways, and they played a variety of roles in regulating plant growth and cold tolerance. The aim of this study was to elucidate the relationship between the developmental stages of winter and spring habit varieties in the wheat seedling period and frost resistance.

## Figures and Tables

**Figure 1 genes-14-00844-f001:**
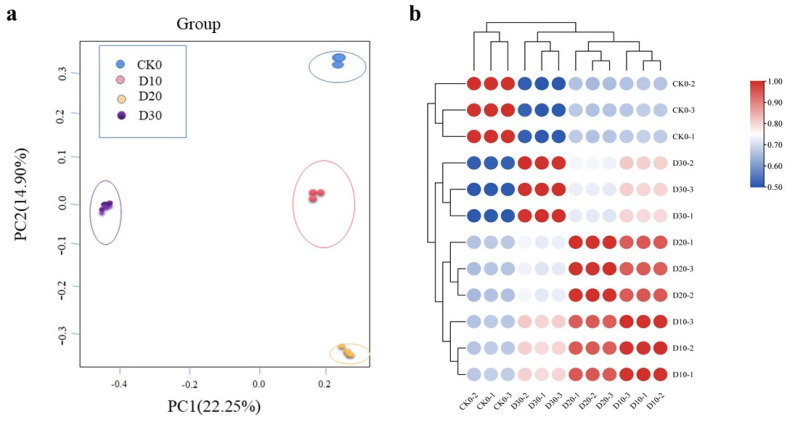
Correlations between the four samples. (**a**) PCA map for all four time points. (**b**) PCC map for all four time points.

**Figure 2 genes-14-00844-f002:**
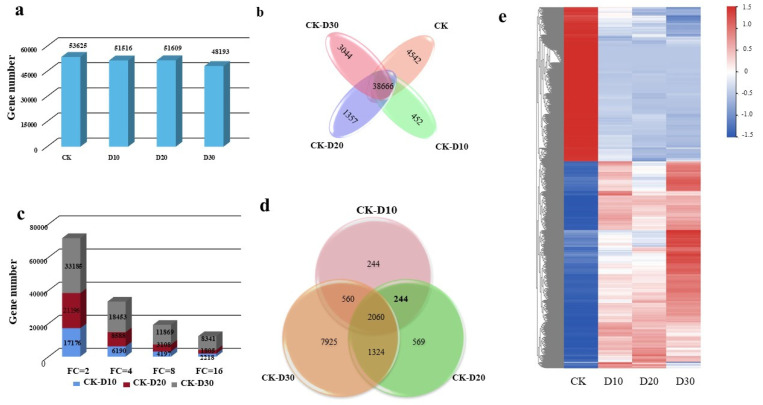
The DEGs’ overview. (**a**) A histogram of the numbers of valid genes in the four samples. (**b**) Venn diagram showing the identification of valid genes in the four samples. (**c**) The number of DEGs with different fold changes. (**d**) Statistical analysis of DEGs among the samples. (**e**) The heat map of DEGs in the four samples.

**Figure 3 genes-14-00844-f003:**
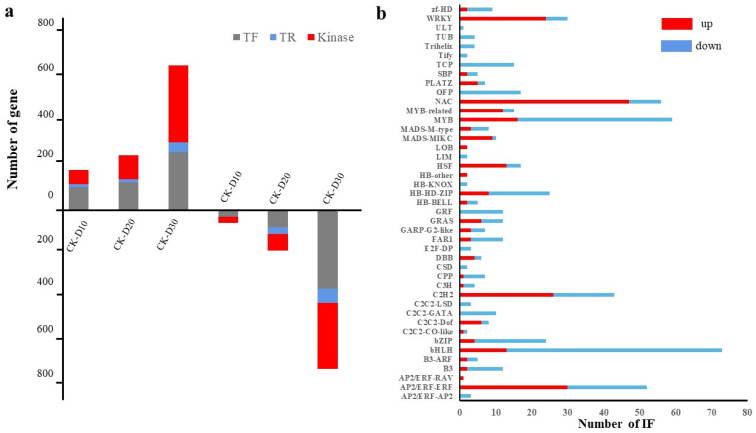
Distribution of transcription factors in DEGs. (**a**) Column diagram of the number of TFs, TR, and kinase in DEGs in different sample pairs. The first three columns represent up-regulated expression, and the last three columns represent down-regulated expression. (**b**) Column diagram of classification of the differentially expressed TFs.

**Figure 4 genes-14-00844-f004:**
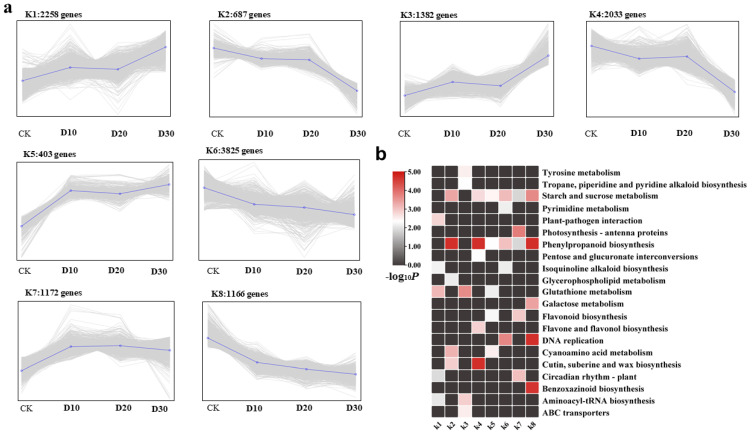
Hierarchical clustering was used to identify clusters of DEGs in the twelve libraries. (**a**) *K*-means clustering of DEGs. (**b**) Top five enrichment pathways of the DEGs in different clusters with reference to KEGG.

**Figure 5 genes-14-00844-f005:**
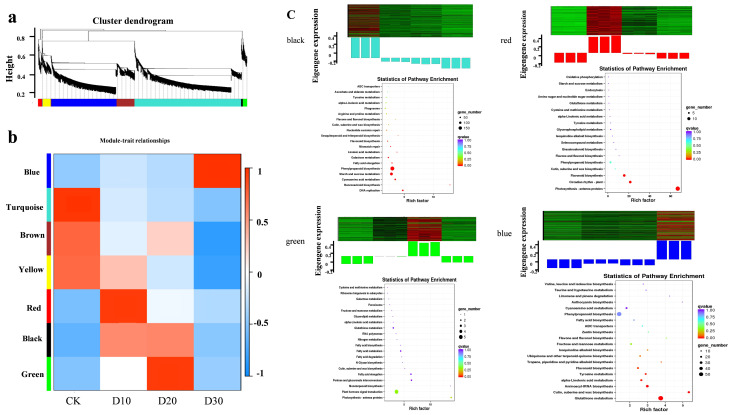
Analysis of co-expression modules of DEGs by WGCNA. (**a**) Co-expression profile of DEGs in this study. (**b**) The correlation coefficient between the different modules and samples. The numbers in each box represent correlation coefficient and standard error. (**c**) The expression level of eigengene and KEGG pathway in different modules.

**Figure 6 genes-14-00844-f006:**
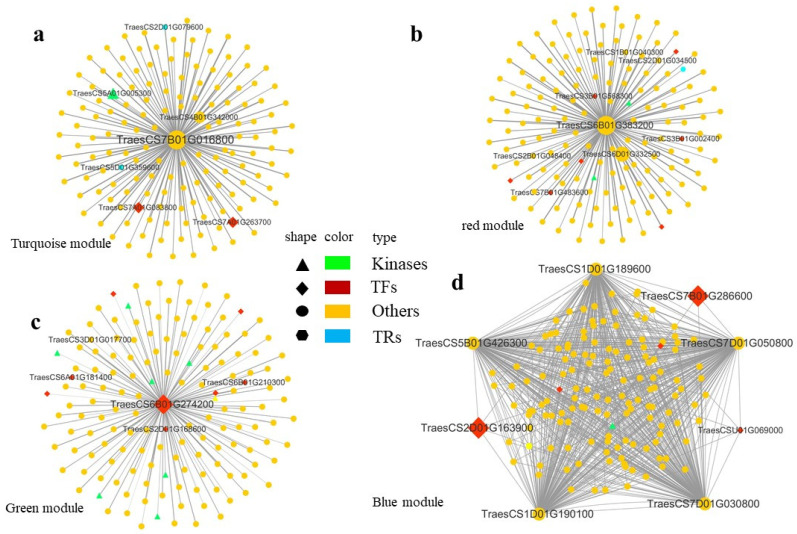
WGCNA was used to establish co-expression networks of DEGs in four significant modules. (**a**) Turquoise module network. (**b**) Red module network. (**c**) Green module network. (**d**) Blue module network.

**Figure 7 genes-14-00844-f007:**
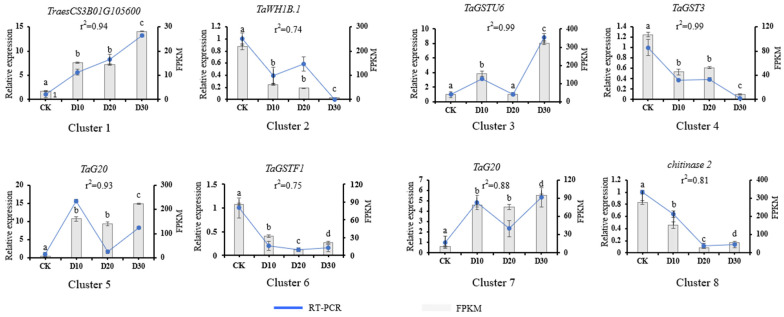
Validation of DEGs was performed using qRT-PCR. A total of eight genes were selected for validation. The x-axis indicates samples in different time spots. The left y-axis indicates relative gene expression levels detected by qRT-PCR. The right y-axis indicates FPKM value. Different letters represent significant differences between the two sets of data.

**Table 1 genes-14-00844-t001:** The main GO enrichment of the DEGs at different treatments.

Sample Pair	Class	Annotation	GO ID	Corrected *p*-Value
CK-D10	Biological process	Nucleosome assembly	GO:0006334	1.14 × 10^−139^
	Cellular component	Nucleosome	GO:0000786	3.04 × 10^−148^
	Molecular function	Protein heterodimerization activity	GO:0046982	7.75 × 10^−107^
CK-D20	Biological process	Nucleosome assembly	GO:0006334	3.78 × 10^−154^
	Cellular component	Nucleosome	GO:0000786	1.10 × 10^−158^
	Molecular function	Protein heterodimerization activity	GO:0046982	1.72 × 10^−113^
CK-D30	Biological process	Nucleosome assembly	GO:0006334	7.55 × 10^−84^
	Cellular component	Nucleosome	GO:0000786	7.02 × 10^−89^
	Molecular function	Protein heterodimerization activity	GO:0046982	4.13 × 10^−56^

**Table 2 genes-14-00844-t002:** Major pathways that the DEGs involved in.

Sample Pair	Pathway	Ko ID	Corrected *p*-Value
CK-D10	Photosynthesis-antenna proteins	ko00196	4.89 × 10^−28^
	Starch and sucrose metabolism	ko00500	2.41 × 10^−17^
	Phenylpropanoid biosynthesis	ko00940	2.19 × 10^−15^
CK-D20	Photosynthesis-antenna proteins	ko00196	1.03 × 10^−25^
	DNA replication	ko03030	1.24 × 10^−18^
	Benzoxazinoid biosynthesis	ko00402	9.09 × 10^−11^
CK-D30	Fatty acid degradation	ko00071	2.23 × 10^−15^
	Porphyrin and chlorophyll metabolism	ko00860	1.25 × 10^−13^
	Starch and sucrose metabolism	ko00500	1.21 × 10^−11^

## Data Availability

Data presented in this study are available in the Appendix A provided with the manuscript. All sequences created for this study have been deposited into the NCBI Short Read Archive database under BioProject PRJNA903114.

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
