# Peer review of "Comprehensive Transcriptome Analysis of Responses during Cold Stress in Wheat (Triticum aestivum L.)"

_genes, 2023, doi:10.3390/genes14040844_

Round 1

Reviewer 1 Report

The manuscript titled 'Comprehensive transcriptome analysis of responses during cold stress in wheat (Tritium aestivum L.)' reports on the molecular  responses of Jing 841 wheat variety under cold stress at the seedling stage. The manuscript gives an insight into wheat gene changes in response to cold stress. The manuscript is of great interest and adds to the progression of abiotic stress studies. This gives the manuscript great relevance in the face of climate change, where certain areas are expected to experience more cold stress than previous seasons.

The introduction, abstract, materials and methods, results and discussion are fairly written and only require minor changes to grammar and the general flow of ideas

In order to improve the quality of the paper, the authors should consider the following:

Abstract

Line 12 - I would advise against beginning a sentence with a digit. You may start the sentence with 'A total of 12,926 differentially expressed genes...' The same comment applies for line 19 as well

Introduction

Line 29 - may you replace 'every year' to 'yearly'

Results

Line 228 - What dos the phrase 'valid genes' mean?

Line 231 and 232 - may the authors rephrase the two sentences to capture what they want to express. As it stands the sentences are difficult to understand.

Figure 9 - May the authors indicate significant differences on the graphs. This can be achieved by the use of letters above the error bars (colour coding may be necessary). As it stands, the reader is unable to deduce statistical differences from the graphs.

Discussion

Line 420 - The sentence is too long and difficult to comprehend. May the authors use two shorter and more precise sentences.

Line 456 - the sentence beginning with WIN1 is incomplete.

Line 459 - This sentence should be combined with the next. 

Line 462 and 464 - is win1 intentionally written in small letters in these statements

Line 463 - replacing 'was' with 'may have been' or any other phrase that does not state it as a fact would be better

Line 468 - replace 'with' with 'of'

Line 484 - correct the sentence to 'not the only'

The authors should thoroughly revise the paper to identify other mistakes that I may have overlooked. 

Reviewer 2 Report

The submitted manuscript “Comprehensive transcriptome analysis of responses during cold stress in wheat (Triticum aestivum L.)” contains a solid set of molecular data related to wheat seedling’s response to low temperature. However the quality of writing needs significant improvement to meet the standard of the presented experiments. There are many confusing sentences and logical flaws, which should be carefully addressed, probably with the help of professional editor.

Just a few examples:

§  In Abstract, lane 15-16: “Starch and sucrose metabolism, glutathione metabolism, and plant hormone signal transduction pathways were found in seedlings”  What is the meaning of that sentence ? Lane 18: “The blue module had the highest correlation coefficient…”, what is the “blue module”, suddenly coming from the blue ?

§  Lane 61: “Cold acclimation and vernalization.” The meaning ?

§  Lane 66-68: “…related to the increased accumulation of several stress- induced proteins including dehydrated alcohol”.  Is alcohol a protein ?  “Plants usually become cold-domesticated within a short period”.  Cold-domesticated ?

§  Lane 77-78:  “…produce hypersensitivity to FT.”  Hypersensitivity to Freezing Tolerance ?

§  Lane 105: “Jing 841 is a winter cultivar with middle maturity…”.    What is “middle maturity”

§  Lane 129-130: “A total amount of 1 μg RNA per sample was used as input material for the RNA sample preparations”.  Maybe preparation of cDNA ?

Other things:

§  Section 3.6 contains a single sentence, I don't think its acceptable. The authors should either provide a more comprehensive description of the obtained result, or move the Fig. 9 to Supplements.  The obtained data on the validation of genes expression should be clearly discussed in relation to the RNAseq results.

§  In the Discussion, authors devote attention to certain pathways and transcription factors. This sections could be improved by adding more targeted and up-to-date references, for example:  

Vogelsang L, Dietz KJ. Plant thiol peroxidases as redox sensors and signal transducers in abiotic stress acclimation. Free Radic Biol Med. 2022 Nov 20;193(Pt 2):764-778. doi: 0.1016/j.freeradbiomed.2022.11.019.

Rajput VD, Harish, Singh RK, Verma KK, Sharma L, Quiroz-Figueroa FR, Meena M, Gour VS, Minkina T, Sushkova S, Mandzhieva S. Recent Developments in Enzymatic Antioxidant Defence Mechanism in Plants with Special Reference to Abiotic Stress. Biology (Basel). 2021 Mar 26;10(4):267. doi: 0.3390/biology10040267.

Bi H, Shi J, Kovalchuk N, Luang S, Bazanova N, Chirkova L, Zhang D, Shavrukov Y, Stepanenko A, Tricker P, Langridge P, Hrmova M, Lopato S, Borisjuk N. Overexpression of the TaSHN1 transcription factor in bread wheat leads to leaf surface modifications, improved drought tolerance, and no yield penalty under controlled growth conditions. Plant Cell Environ. 2018 Nov;41(11):2549-2566. doi: 10.1111/pce.13339.

Reviewer 3 Report

The research article entitled “Comprehensive transcriptome analysis of responses during cold stress in wheat (Triticum aestivum L.)” by the authors provides well-structured answers to one of a major abiotic stress i.e. cold stress.

Overall, they address an important question, pertaining to the role of cold stress in wheat at the seedling stage. More importantly it talks about various transcription factors and glutamate metabolism pathway genes that has potential significance for improving freezing tolerance in wheat. The MS requires minor corrections that will help immensely in shaping the manuscript

Reviewers' comments to the author:

Title: Comprehensive transcriptome analysis of responses during 2 cold stress in wheat (Triticum aestivum L.)

Reviewer #1:

The topic of this research article is well chosen, useful and innovative. However, it needs to be improved and thoroughly revised. Greater attention should be paid to the following points:

1.     Line 11,26, 28, 35, 39, 41, 42, 76, 91-93 scientific name of the plant species should be in italics

2.     Kindly restructure the sentence in line 60-61

3.     The name of the genes should be in italics.  Also, check and correct few minor formatting-related mistakes carefully in the manuscript

4.     As the topic of the research article is timely and important, I suggest that the authors write the objective of the research in details in the introduction.

5.     Line 232 doesn’t make any sentence; authors are request to restructure it.

6.     Could you kindly briefly explain the “eight DEGs for transcription validation by qRT-PCR” in section 3.6.

7.     Furthermore, there were still some typos in the MS and the authors are requested to read their MS carefully before submitting.

Reviewer 4 Report

An up-to-date research, methodologically very well covered
